# Antigene *MYCN* Silencing by BGA002 Inhibits SCLC Progression Blocking mTOR Pathway and Overcomes Multidrug Resistance

**DOI:** 10.3390/cancers15030990

**Published:** 2023-02-03

**Authors:** Sonia Bortolotti, Silvia Angelucci, Luca Montemurro, Damiano Bartolucci, Salvatore Raieli, Silvia Lampis, Camilla Amadesi, Annalisa Scardovi, Giammario Nieddu, Lucia Cerisoli, Francesca Paganelli, Francesca Chiarini, Gabriella Teti, Mirella Falconi, Andrea Pession, Patrizia Hrelia, Roberto Tonelli

**Affiliations:** 1BIOGENERA SpA, R&D Department, 40064 Bologna, Italy; 2Pediatric Oncology and Hematology Unit, IRCCS, University Hospital of Bologna, 40138 Bologna, Italy; 3Oncodesign SA, 21079 Dijon, France; 4Bambino Gesu Children’s Hospital, IRCCS, Research Laboratories-Oncohematology Department, 00165 Rome, Italy; 5Department of Biomedical and Neuromotor Sciences, University of Bologna, 40126 Bologna, Italy; 6CNR Institute of Molecular Genetics “Luigi Luca Cavalli-Sforza”, Unit of Bologna, 40129 Bologna, Italy; 7Department of Bio-medical, Metabolic and Neural Sciences, Section of Human Morphology, University of Modena and Reggio Emilia, 41125 Modena, Italy; 8IRCCS, Pediatric Unit, University Hospital of Bologna, 40138 Bologna, Italy; 9Department of Pharmacy and Biotechnology, University of Bologna, 40126 Bologna, Italy

**Keywords:** small-cell lung cancer, MYCN, targeted therapy, mTOR

## Abstract

**Simple Summary:**

Small-cell lung cancer (SCLC) is the most aggressive form of lung cancer, is mostly associated with smoking and has a low survival rate. Among the different alterations in genes identified in SCLC, those related to the *MYC* family have emerged as highly relevant, alterations *MYCN* particularly define an aggressive and immunotherapy resistant SCLC subgroup. Due to its highly restricted expression in normal cells, *MYCN* is an ideal target for precision medicine, though it is difficult to target with traditional approaches. We propose an innovative approach to target *MYCN* in SCLC by an *MYCN*-specific expression inhibition at the level of gene transcription through an antigene oligonucleotide (BGA002). BGA002 exerted a potent and specific *MYCN* silencing in *MYCN*-expressing SCLC, leading to reversion of specific pathways and induction of cell death. Moreover, systemic administration of BGA002 inhibited tumor progression and significantly increased survival in SCLC mouse models, while also overcoming multidrug resistance.

**Abstract:**

Small-cell lung cancer (SCLC) is the most aggressive lung cancer type, and is associated with smoking, low survival rate due to high vascularization, metastasis and drug resistance. Alterations in *MYC* family members are biomarkers of poor prognosis for a large number of SCLC. In particular, *MYCN* alterations define SCLC cases with immunotherapy failure. *MYCN* has a highly restricted pattern of expression in normal cells and is an ideal target for cancer therapy but is undruggable by traditional approaches. We propose an innovative approach to *MYCN* inhibition by an *MYCN*-specific antigene—PNA oligonucleotide (BGA002)—as a new precision medicine for *MYCN*-related SCLC. We found that BGA002 profoundly and specifically inhibited *MYCN* expression in SCLC cells, leading to cell-growth inhibition and apoptosis, while also overcoming multidrug resistance. These effects are driven by mTOR pathway block in concomitance with autophagy reactivation, thus avoiding the side effects of targeting mTOR in healthy cells. Moreover, we identified an *MYCN*-related SCLC gene signature comprehending *CNTFR*, *DLX5* and *TNFAIP3*, that was reverted by BGA002. Finally, systemic treatment with BGA002 significantly increased survival in *MYCN*-amplified SCLC mouse models, including in a multidrug-resistant model in which tumor vascularization was also eliminated. These findings warrant the clinical testing of BGA002 in *MYCN*-related SCLC.

## 1. Introduction

Lung cancer (LC) is the leading cause of cancer deaths worldwide and small-cell lung cancer (SCLC) accounts for approximately 15–20% of all lung neoplasms [1,2]. SCLC is the most aggressive form of LC: median overall life expectancy is about 12 months and long-term survival is rare [3]. SCLC is strongly associated with tobacco smoking: more than 90% of patients are elderly and are current or past heavy smokers, with the risk rising with increasing duration and intensity of smoking [1,4]. SCLC is characterized by rapid growth, high vascularity, and early metastatic dissemination, mainly in brain, liver, adrenal glands, bones and bone marrow [5,6].

Although there has been decades of research and multiple clinical trials have been carried out [7,8], little progress has been made and therapeutic options for SCLC patients are still limited [6,9,10]. First-line treatment is etoposide–platinum-based chemotherapy [11], with consideration of prophylactic cranial irradiation (PCI) in order to reduce the risk of brain metastases [12,13]. Patients demonstrate a good initial response to treatment, but drug resistance soon arises and the vast majority of patients experience recurrence [14,15,16].

Currently the FDA has approved topotecan and lurbinectedin, as second-line treatments [17,18,19]. Recently, immune checkpoint inhibitors have been approved in combination with chemotherapy agents for metastatic SCLC, however, the clinical benefits were limited, and only a small number of patients have benefited from these immune-based therapies [20,21,22]. Given these premises, the development of novel targeted therapies for SCLC remains an unmet medical need.

Numerous genetic alterations have been identified in this lung cancer type, including proto-oncogene activation and loss or inactivation of tumor suppressor genes [4,23,24].

Among these genomic alterations, amplification and overexpression of *MYC* family genes occur in 20–40% of SCLC cases, in a mutually exclusive manner [6,25]. The *MYC* family of oncogenes and its extended protein network is involved in the regulation of several processes such as cell growth, cell cycle, proliferation, differentiation, survival and apoptosis, therefore mutations in these genes are frequently associated with poor prognosis and shortened survival in many types of cancers, including SCLC [9,26,27,28]. Interestingly, SCLC produces the only form of tumor in which all the *MYC* family genes (*MYC*, *MYCN* and *MYCL*) are found to be altered. Despite their similarities, *MYC* members characterize different subsets of SCLC. *MYC* drives a non-neuroendocrine (non-NE) subset of SCLC, which has a mesenchymal-like phenotype [29], *MYCL,* on the other hand, is highly expressed in neuroendocrine (NE) SCLC [30]. An elevated *MYCN* expression has been found in a specific subset of NE SCLC—defined SCLC-N—which presents immune-escape and resistance to immunotherapies [29,31].

Physiologically, each member of the *MYC* family exhibits a different pattern of expression regarding timing and lineage [26,29]. While *MYC* is ubiquitously and highly expressed in cells throughout development and in adult tissues, *MYCL* expression is predominantly restricted to both neonatal and adult lung tissue; in contrast *MYCN* expression is tumor specific and is not usually expressed in human adult tissues [26,27,32]. Thus, the inhibition of *MYCN* represents a new relevant cancer treatment, specific for *MYCN*-driven human tumors, without affecting healthy cells [25,32]. Numerous approaches have been proposed for *MYCN* inhibition, including blocking its heterodimerization with *MAX* or silencing it indirectly; however, all these strategies have largely failed. Thus, *MYCN* has long been considered an undruggable target by traditional drug discovery approaches [32]. In this context, we have previously shown that a novel strategy using an *MYCN*-specific antigene peptide nucleic acid (agPNA) oligonucleotide (called BGA002) is effectively able to silence *MYCN* expression by acting directly at the DNA level in neuroblastoma (NB) [33,34,35,36].

In this work, we show that *MYCN* inhibition by BGA002 is specific and highly effective in the context of *MYCN*-related SCLC. Silencing of *MYCN* induces alterations in the expression of downstream genes and results in mTOR signaling blockade and autophagy activation. Moreover, BGA002 is effective in vivo as well, significantly improving survival in *MYCN*-amplified (MNA) SCLC mouse models (including one that is multidrug resistant).

## 2. Materials and Methods

### 2.1. Cell Lines and Cell Culture

All cell lines were stored in liquid nitrogen and kept in culture for a maximum of 30 days and less than seven passages from the time they were obtained. The average number of passages for each cell line used in this study is three. Cell lines were verified to be negative for the presence of *Mycoplasma* every three months by a PCR-based method with the LookOut Mycoplasma PCR detection kit (Sigma-Aldrich, Burlington, MA, USA) using the manufacturer’s instructions. NCI-H69, H69AR, NCI-H526, DMS 79 and NCI-H510A were purchased from ATCC, HEK293 was purchased from DSMZ, GLC-14 was kindly gifted by Prof. Elisabeth G E de Vries (Groningen, The Netherlands) and NCI-N592 was kindly gifted by Dr. Silvano Ferrini (Genoa, Italy). Cells were maintained in standard culture conditions, according to seller until experimental procedures were required. The list of cell lines used in this study with additional details is available as part of the Appendix A.

### 2.2. Cell Line Treatment

BGA002 was produced by Biogenera SpA. The PNA-peptide was prepared by the chemistry department and delivered to the biology department after purification and dilution. The PNA was freshly produced and used or stored at 4 °C. PNA was designed and prepared according to previously published studies [33,34]. Cell lines were expanded in RPMI-1640, with 10% FBS. Cells were counted using a Burker’s chamber and resuspended with RPMI-1640, with 0.5% FBS. For PNA-peptide treatment, 1 × 10^5^ cells were plated in a 24-well, flat-bottom plate for RNA-extraction. For cell viability assays 10 × 103 cells were seeded in a 96-well flat-bottom plate. SCLC cell lines were treated with increasing concentrations ranging from 0.16 μM to 20 μM. siRNA for CNTFR was mixed with Lipofectamine (Invitrogen) and then diluted in RPMI-1640. An amount of 1 × 10^5^ cells were plated in a 24-well plate and incubated with siRNA (ranging from 0.6 nM to 20 nM). After 6 h of treatment, 9.5% of FBS was added to cells.

### 2.3. Quantitative Real-Time PCR

RNA extraction, retro-transcription and real-time PCR were performed as previously described [35]. The list of primers used in this study is given in Appendix A.

### 2.4. Cell Viability Assay and Western Blot Analysis

Cell viability assays were performed as previously described [35]. Cell viability assays were performed using CellTiter-Glo Viability Assay^®^ kit Promega. Western Blot analysis was conducted using standard methods [37]. Briefly, cells were lysed in RIPA lysis buffer with protease inhibitor (Thermo Fisher Scientific, Waltham, MA, USA), and then sonicated. Protein fraction was collected by centrifugation at 13,000× g, 4 °C, for 10 min. A total of 25 μg of proteins were loaded and separated by SDS–PAGE using Criterion TGX polyacrylamide gels (Bio-Rad, Hercules, CA, USA) and blotted into a nitrocellulose membrane (Bio-Rad, Hercules, CA, USA). We tested the expression of proteins of interest using the following antibodies from Cell Signaling Technology (Danvers, MA, USA): anti-Phospho-Akt (Ser473) (#4060) 1:1000; anti-Akt (#9272) 1:1000; anti-Phospho-p70 S6 Kinase (Thr389) (#9206) 1:1000; anti-p70 S6 Kinase (#9202) 1:1000; anti-Phospho-S6 Ribosomal Protein (Ser235/236) (#4858) 1:1000; anti-S6 Ribosomal Protein (#2217) 1:1000; anti-Phospho-4E-BP1 (Thr37/46) (#2855) 1:1000; anti-4E-BP1 (#9452) 1:1000; and anti-glyceraldehyde 3-phosphate dehydrogenase (GAPDH) (#5174) 1:1000. We also employed anti-N-Myc (sc-53,993) 1:800 from Santa Cruz Biotechnology (Dallas, TX, USA). Bands were detected using the Cyanagen Westar ECL Western blotting detection reagent (Cyanagen Bologna, Italy). Images were captured by ChemiDoc-It2 Imaging System and analyzed with the Vision Works LS Software (UVP, LLC, Upland, CA, USA).

### 2.5. Apoptosis Analysis

The NCI-N592, H69AR and NCI-H510A cells were treated as described above. Cells were stained with an Annexin V/FLUOS Staining Kit (F. Hoffmann-LaRoche AG, Basel, Switzerland) according to the manufacturer’s instructions. The cell samples were analyzed via CytoFLEX flow cytometer (Beckman Coulter Inc., Brea, CA, USA). The results were analyzed using FlowJo software (Tree Star Inc. Ashland, OR, USA).

### 2.6. Transmission Electron Microscopy

The NCI-N592, H69AR and NCI-H69 cells were seeded in a 6-well plate. After 24 h, cells were treated with NaCl 0.9%, BGA002 10 μM in low FBS culture medium. After 6 h, up to 10% of FBS was added. After 48 h of treatment, the cells were fixed. Cells were removed from well, washed in warmed PBS then fixed with 2.5% (*v*/*v*) glutaraldehyde in cacodylate buffer 0.1 mol/L for 2 h at 4 °C. Cells were then post fixed with a solution of 1% osmium tetroxide in 0.1 mol/L cacodylate buffer and embedded in epoxy resins after a graded-acetone serial dehydration step. Ultrathin slices of 100 nm were stained by 3% uranyl acetate (*v*/*v*) in 50% ethanol solution and 3% lead citrate (*v*/*v*) in water, and then observed with transmission electron microscope CM100 Philips (FEI Company, Eindhoven, The Netherlands) at an accelerating voltage of 80 kV. Images were recorded by Megaview III digital camera (FEI Company, Eindhoven, The Netherlands).

### 2.7. Lysosomes Distribution Analysis

The NCI-H69, H69AR and NCI-N592 cell lines were seeded in a Nunc Lab-Tak Flask on Slide for live staining. Treatment was administered 48 h before acquisition. Lyso-Tracker was added and the cells were incubated for 45 min at 37 °C, at 5% CO_2_. For each condition, z-stacks (at a 200 nm interplane distance) were acquired using a Nikon Ti2-E microscope (Nikon, Tokyo, Japan). Images were elaborated using the Fiji plugin in ImageJ software. Images containing lysosome were modified for background reduction. Each image was duplicated then a maximum with radius 3 was applied, then gaussian blur was applied with sigma 100 μm in one image replicate. In the other replicates a subtract backgrounds was applied with rolling ball methods with radii of 100 μm. The first image was then subtracted to the second one using an image calculator for subtracting, a gaussian blur of 1 μm was then applied. Images were then binarized using threshold, this method was selected as minimum before the images are watershed. Lysosomes were then analyzed using Analyze>Analyze Particles, with the lower value size set to 0.1 μm^2^.

### 2.8. Xenograft Ectotopic SCLC Mouse Models

NCI-H69-Luc and H69AR-Luc were prepared as described previously [35]. All experiments were approved by the Scientific Ethical Committee of Bologna University (protocol no. 564/2018-PR). Six-week-old mice (NOD/SCID CB17; both sexes) were inoculated with 10 × 10^6^ NCI-H69-Luc or H69AR-Luc in the dorso-posterio-lateral position. Mice were sedated with isoflurane prior to the injection. Luminescence was used to monitor the growth of tumors (D-Luciferin was administered via intraperitoneal injection, and luminescence was monitored using the UviTec Imaging System (Cleaver Scientific, Ltd., Rugby, UK). Treatment administration began after a predefined starting point during bioluminescent acquisition and was conducted daily for 28 days with a subcutaneous injection of 100 μL of vehicle, 50 mg/kg/day of BGA002 and 100 mg/kg/day of BGA002. Animals were monitored until they reached the endpoint (20 mm linear tumor or 60 days post treatment). Tumor size and volume was calculated using a caliper. After reaching the endpoint, the mice were sacrificed. The tumors were removed, measured, weighed, and fixed in 4% formalin. Immunohistochemistry was conducted as described previously [35].

### 2.9. Statistical Analysis

All data were analyzed using GraphPad Prism 8 software or with R software version 3.5 or Python software version 3.7. The different analyses and tests were specifically designed for each experiment.

## 3. Results

### 3.1. BGA002, a Specific MYCN Antigene Oligonucleotide, Strongly Inhibits MYCN Tumorigenic Activity in SCLC Cells

We have previously demonstrated that BGA002, as a single agent, was able to specifically inhibit *MYCN* expression in neuroblastoma cell lines [35]; furthermore, in combination with 13-cis RA, BGA002 has shown an improvement in *MYCN* transcriptional inhibition [36]. This inhibition led to reduced cell viability due to apoptotic phenomena in vitro, and antitumor activity in mouse model in vivo [35,36].

Given these premises, we investigated whether BGA002 is able to inhibit *MYCN* expression in another *MYCN*-driven tumor, SCLC.

First, we selected a panel of six SCLC cell lines to determine BGA002 in vitro activity: five cell lines were *MYCN* amplified (MNA) and one was characterized by *MYCN* overexpression, but not amplification (not-MNA). As expected, all the cells expressed high levels of *MYCN* mRNA (Appendix A). One MNA–SCLC cell line (H69AR) was also multidrug resistant.

BGA002 treatment induces a marked dose-dependent reduction of *MYCN* transcript, followed by a diminishment in N-Myc protein levels and accompanied by growth inhibition in SCLC cells (Figure 1A–C and Appendix A). In all cell lines EC_50_ values are <1 µM for *MYCN* mRNA and range from 2.50 µM to 5.50 µM for cell viability (Appendix A).

BGA002 effect was specific and did not influence cell viability in the *MYCN*-unexpressed HEK293 cell line (Appendix A). This was confirmed by another experiment on a different *MYCN*-unexpressed cell line, the NCI-H510A (Appendix A). Indeed, BGA002 did not induce apoptosis in this cell line, whereas there was a dose-dependent increase in apoptosis in MNA–SCLC cells, after 48 h of treatment (Figure 1D).

The induction of apoptosis following treatment was also confirmed by TEM analysis, wherein we could observe the first signs of nuclear chromatin condensation in NCI-H69 and H69AR cells in early apoptosis, while cells in late apoptosis showed clear semilunar chromatin condensation, and there were no more recognizable cytoplasmic structures (Appendix A).

### 3.2. BGA002 Treatment Increases Lysosome Number and Modifies Their Distribution in SCLC Cells

BGA002 was able to induce a decrease in cell viability, caused by apoptosis activation. In this respect, as reported previously, autophagy triggered by lysosome activation seems to be a major pathway involved in defining cell death [36]. Interestingly, lysosomes have also been reported to be important features of autophagy and drug resistance-related processes in LC [38,39], and their modulation has been demonstrated to be able to reactivate tumor response to treatment and therapy in both LC [40,41,42] and SCLC [43]. We wondered if BGA002 was able to modulate lysosomes activation in SCLC cell lines. Electron microscopy analysis showed an effect of BGA002 on lysosome physiology and organization. In fact, lysosome and lysosomal-related structures, were found in cells after treatment, but were not evident in the control (Figure 2A). Interestingly, lipid degradation, normally associated with the autophagy process, has been found only in NCI-H69 and H69AR cell lines, but not in NCI-N592. Lysosomes analysis performed with fluorescent tracker (Appendix A) confirmed the previous results. In fact, BGA002 treatment showed an increase in lysosomes number and a modified distribution in all MNA SCLC cells (Figure 2B). While mean diameters were unaltered, treated cells showed a particular shift from lower diameters (<1 µm) to higher (Figure 2B), highlighting a major lysosomal activity [44]. Raw data are reported in Appendix A.

### 3.3. MYCN Inhibition Blocks mTOR Complex in SCLC Cells

Data in the literature have demonstrated that *MYCN* amplification led to activation of many downstream pathways including PI3K/AKT/mTOR, which is a master regulator of cell growth, proliferation and metabolism [45,46]. Genetic alterations in mTOR signaling have been implicated in metabolic disorders, neurodegeneration, ageing and in various cancer types [45,47]; in particular, dysregulations of mTOR pathway have been detected in 36% of patients with SCLC and have been related to radiation and chemotherapy resistance [47,48]. Krencz and colleagues have highlighted the relevance of mTOR pathway in the progression and metastasis formation of SCLC [49].

The PI3K/Akt/mTOR pathway has become an important therapeutic target for SCLC [48,50] because several studies have shown that its inhibition results in reduced growth, promotion of apoptosis and enhanced sensitivity to cisplatin/etoposide in both SCLC cells and PDX models [50,51,52]. Despite promising preclinical results, mTOR inhibitors that reached clinical trials failed to improve survival of SCLC patients [50,51].

Here we report that the expression of key genes belonging to mTOR signaling (*SLC7A5*, *MLST8* and *EIF4EBP1*) correlated with *MYCN* expression in SCLC patients (Figure 3A). Intriguingly, the same was not observed for *MYC* or *MYCL* (Appendix A). Thus, we have hypothesized the possibility of down-regulating mTOR complex via *MYCN* inhibition for the treatment of *MYCN*-expressing SCLC.

We found that BGA002 treatment led to inhibition of mTOR genes in all *MYCN*-related SCLC cell lines (Figure 3B). mTOR pathway activity was also evaluated through protein phosphorylation. An analysis showed a reduction of Akt, P70S6K, S6RP and 4E-BP1 phosphorylation after treatment in MNA–SCLC cells (NCI-N592) (Figure 3C,D). Our findings demonstrate that *MYCN* inhibition, mediated by BGA002, strongly inhibited mTOR pathway in *MYCN*-expressing SCLC.

### 3.4. BGA002 Regulates Multiple Cancer Related Pathways in SCLC Cells

Gene analysis on SCLC cell lines not only confirmed mTOR pathway as affected by *MYCN* inhibition but highlighted other genes involved in tumorigenesis as a potential key regulator of BGA002’s mechanism of action. (Figure 4A).

CLCF1-CNTFR signaling is emerging as a target for specific therapy for non-small-cell lung cancer (NSCLC) [53,54,55], while nothing has previously been evaluated for SCLC. Interestingly, we found that *MYCN* inhibition by BGA002 resulted in a strong down-regulation of *CNTFR* in all *MYCN*-related SCLC cell lines, independent of MNA status or chemo-resistance (Figure 4A). Moreover, in a panel of several different tumor types, *CNTFR* showed low mRNA expression in the majority of the tumors, while in NB (which is known for its connection with MNA and *MYCN* overexpression) and in SCLC it showed the highest expression (Figure 4B). Taken together this information led us to evaluate whether *MYCN* inhibition by BGA002 can exert its role against *MYCN*-related SCLC, through *CNTFR* inhibition. We found that specific inhibition of *CNTFR* can efficiently reduce cell viability in MNA–SCLC cells (NCI-N592), accounting for a large percentage of the effect mediated by BGA002 (Figure 4C), confirming the importance of *CNTFR*. To further extend these findings, we evaluated whether *MYCN* inhibition can affect *CNTFR* expression in NB. *CNTFR* mRNA expression in MNA–NB cells (Kelly) was reduced after BGA002 administration (Figure 4D), which also extended the relationship between *MYCN* and *CNTFR* in another *MYCN*-related tumor.

*DLX5* is another gene that is significantly down-regulated upon inhibition of *MYCN* expression by BGA002 in our analysis in *MYCN*-related SCLC (Figure 4A). This gene has been reported in association with tumor size and poor prognosis and cell proliferation in NSCLC [56]. *DLX5* emerged from a panel of different tumors, among which it showed the highest expression in SCLC (Appendix A).

Gene expression analysis in patients conducted on both SCLC and NB confirmed our findings regarding *DLX5* correlation with *MYCN* (Appendix A). Over-expression of *DLX5* in SCLC resulted as an exclusive feature of *MYCN* over-expression in patients, in comparison with *MYC* and *MYCL* expressing patients (Appendix A). We found a connection between *DLX5* and MNA in NB, which ranked among the tumors with the highest *DLX5* expression (Appendix A). *DLX5* expression in NB was also responsible for reduced survival probability (Appendix A). Moreover, *MYCN* inhibition led to a consistent reduction of *DLX5* in MNA-NB (Appendix A).

We found that BGA002 was able to induce upregulation of genes in *MYCN*-related SCLC, as it was for *TNFAIP3* (Figure 4A). *TNFAIP3* is known to directly down-regulate the pro-inflammatory NF-κB pathway [57,58]. Indeed, we also found that two relevant genes (NFKB1 and RELA) involved in this pathway, were down-regulated after BGA002 treatment in *MYCN*-related SCLC (Figure 4A).

We further confirmed our findings on *TNFAIP3* implication in SCLC by our analysis on a panel of different tumor types, where *TNFAIP3* showed the second lowest expression in SCLC (while the lowest expression was found in NB) (Appendix A).

Within the MYC family, gene expression analysis in SCLC patients showed a specific association between only *MYCN* and *TNFAIP3*, while it did not with *MYC* and *MYCL*. In particular, high *MYCN* expression appeared to be correlated with low levels of *TNFAIP3* in SCLC (Appendix A). Gene expression analysis performed in MNA-NB patients also confirmed our findings. MNA-NB patients showed a lower level of *TNFAIP3* compared with non-MNA-NB, and survival probability appeared higher in patients where *TNFAIP3* expression was maintained (Appendix A). Interestingly, BGA002 was also able to restore *TNFAIP3* mRNA expression in MNA-NB cells (Appendix A).

We found many other relevant genes regulated by BGA002 in *MYCN*-related SCLC, including genes involved in metabolism (*G6PD, G6PC3, IGF2*) [59,60,61], and stress granules (*ATXN2, ATXN2L, DDX1* and *SAMD4B*) [62,63,64] (Figure 4A).

### 3.5. BGA002 Improves Survival in MNA-SCLC Xenograft Mouse Models, Independently of Multidrug Resistance

Finally, we evaluated the in vivo antitumor activity of BGA002 in two different xenograft models of MNA-SCLC: NCI-H69-Luc and multidrug-resistant H69AR-Luc. The systemic administration (subcutaneously (SC), daily for 28 days) of BGA002 was able to reduce tumor growth in comparison to the vehicle in both models (Figure 5A).

BGA002 treatment also resulted in significant survival augmentation in both MNA-SCLC models (Figure 5B). The 50 mg/kg/day treatment induced an increase in survival in NCI-H69-Luc, but not in the multidrug-resistant H69AR-Luc. However, by increasing the BGA002 dosage to 100 mg/kg/day, we overcame this phenomenon, which also led to significant survival augmentation in the H69AR-Luc model (Figure 5B).

Histological analysis of tumors, comparing animals treated with vehicle or with 50 mg/kg/day of BGA002, revealed a relevant tumor vascularisation in the multidrug-resistant model that was eliminated after treatment (Figure 5C). Moreover, immunohistochemistry analysis after BGA002 treatment showed a consistent reduction of N-Myc protein expression and a decrease of Ki-67, in both models (Figure 5C). Finally, no animal used in the study showed sign of toxicity or weight loss (Appendix A) after administration of BGA002 at 50 or 100 mg/kg/day.

## 4. Discussion

Small-cell lung cancer still represents a challenging pathology. The high aggressiveness and the lack of proper therapeutic options are strongly implicated with poor prognosis and low survival rates [65]. Conventional and new therapies have largely failed to improve SCLC outcomes [6], which have shown a consistent lack of response to immunotherapies [22]. Thus, innovative targeted therapy approaches have been sought to improve this scenario.

Interestingly, the *MYC* family of oncogenes is frequently mutated in SCLC [6,26] and is related to several aggressive features and poor outcomes [66,67,68]. In recent studies, MYC family was also correlated with different SCLC subtypes [29,30,69]. *MYC* and *MYCL* appeared to specifically express in non-neuronal/mesenchymal-like and neuronal SCLC, respectively [29]. Moreover, classification based on neuroendocrine proteins have shown that *MYCN* is strongly associated with the SCLC-N subtype. Interestingly, SCLC-N is the most unresponsive to immunotherapies [31]. As we have previously reported, *MYCN* expression is correlated with immune repression in different tumors, such as SCLC, NB, rhabdomyosarcoma (RMS), retinoblastoma (RB), Wilms’ tumor, T-cell acute lymphoblastic leukemia (T-ALL) and acute myeloid leukemia (ALM) [70]. For this reason, specific inhibition of *MYCN* in *MYCN*-related tumors could restore the immune response, as in the case of NB after treatment by BGA002 [70].

In this context, the *MYCN*-specific antigene oligonucleotide BGA002 proved to have high specificity and efficacy in *MYCN*-related NB [35,36], thus also representing a novel precision medicine approach for *MYCN*-related SCLC treatment. In this study, BGA002 showed potent and specific dose-dependent downregulation of *MYCN*, at both the mRNA and protein levels, in MNA and *MYCN*-expressing SCLC cell lines, including multi-drug resistant cells.

We further investigated other mechanisms modulating *MYCN*-related SCLC response after BGA002 treatment. We found high lysosomal rearrangements under *MYCN* inhibition stimuli, leading to reactivation of phagocytosis; as reported also for NB [36]. Following these findings, we analyzed the mTOR pathway, understanding its correlation with phagocytosis process, and found a consistent reduction in mTOR pathway activity. Gene expression analysis showed that BGA002 inhibited many genes correlated with mTOR pathway activation in *MYCN*-related SCLC, leading also to the loss of phosphorylation of mTOR proteins. The *MYCN* silencing caused by BGA002 constitutes an innovative precision medicine approach, enabling the inhibition of mTOR to occur only in cancer cells, and leaving normal cells unaffected. Consequently, it avoids the side effects induced by traditional mTOR inhibitors such as rapamycin and its analogs (rapalogs), that also interfere in healthy cells [71,72].

In our finding, other genes, such as *CNTFR*, *DLX5* and *TNFAIP3* emerged as centrals, for the first time, in SCLC. The *CNTFR* gene was reported as a new marker for NSCLC assessment and a potential therapeutic target [53], particularly for its correlation with tumor growth, differentiation modulation and pro-inflammatory cytokines stimulation [54,55]. Remarkably, we found that, among different tumors, *CNTFR* showed the highest expression in SCLC and NB, which are two *MYCN*-related neuroectodermal tumors [73,74]. Moreover, *MYCN* inhibition by BGA002 resulted in a strong down-regulation of *CNTFR* in both *MYCN*-related SCLC and NB. Finally, we also found that *CNTFR* specific inhibition can efficiently reduce cell viability in MNA-SCLC cells, confirming the importance of *CNTFR* in this tumor subtype.

*DLX5* also emerged as extremely relevant in SCLC. Among different tumors, *DLX5* showed the highest expression in SCLC (with NB also among showing higher expression). This gene also appeared to specifically correlate with *MYCN* (but not with *MYC* or *MYCL*) expression in SCLC patients. Interestingly, here we report that *DLX5* is also highly correlated with MNA and unfavorable prognosis in a large cohort of MNA-NB patients. Moreover, *DLX5* was downregulated by BGA002 in both *MYCN*-related SCLC and MNA-NB. Furthermore, our group previously identified genes linked to differentiation that were differentially active in two NB cohorts [36]. Among these genes, *DLX5* was mainly associated with the advanced pathology stage in NB patients [36]. All of these findings strongly increase the interest for *DLX5* as a potential therapeutic target.

Intriguingly, we found that *TNFAIP3* showed the lowest expression in NB and SCLC, among different tumors. This gene also emerged as inversely correlated with *MYCN* (but not to *MYC* or *MYCL*) expression in SCLC patients, and in a similar manner, we found that in NB patients, it is inversely correlated with MNA and is positively linked to survival probability. Remarkably, we also showed that *TNFAIP3* expression can be restored by BGA002 in both *MYCN*-expressing SCLC and NB cells. *TNFAIP3* is known to directly down-regulate the pro-inflammatory NF-κB pathway [57,58], and it was also found to be a signature gene of M1 macrophages [75].

Interestingly, we previously found that *MYCN* expression was inversely correlated with immune response pathways in different tumors, including SCLC and NB. Specifically, we previously demonstrated that *MYCN* amplification in NB showed anti-correlation with Th1/M1 immunity and a correlation with pro-tumoral Th2/M2, while BGA002 was able to restore NK susceptibility in MNA-NB [70]. Taken together, this information indicates that the upregulation of *TNFAIP3* induced by BGA002 could be involved in the inhibition of the inflammation and in the restoration of the Th1/M1 susceptibility in *MYCN*-related SCLC and NB, leading to anti-tumor immunity restoration [75]. In the context of SCLC, this phenomenon is of particular relevance for the SCLC-N (the *MYCN*-related SCLC subtype) that is characterized by immune escape and immunotherapy failure [31].

While SCLC and NB occur at different ages (adult and pediatric patients, respectively) and originate in different sites (lung and adrenal medulla, respectively), they share several similarities. These overlapping features include their neuroectodermal provenience, their clinical profile—including their aggressiveness and poor prognoses—drug resistance, metastasis (particularly in the bone marrow), immune escape and immunotherapy failure.

Our findings reveal that the expression of *MYCN* in the two tumors also drives similarities in the pathways (such as mTOR activation) and in the deregulation of genes (in particular upregulation of *CNTFR* and *DLX5* and downregulation of *TNFAIP3*). Additionally, we have shown that *MYCN* inhibition by BGA002 has the ability to revert this in the *MYCN*-related SCLC and NB, leading to potent and specific anti-tumor activity.

Thus, our data offer another example of a precision medicine approach, in which different tumors are unified in their targeted therapy treatment by their common driver cancer gene (*MYCN*).

Moreover, while drug resistance imposes a failure that reversed the initial response to chemotherapy in patients with SCLC, here we have shown that anti-tumor activity of BGA002 against *MYCN*-related SCLC occurs independently of multidrug resistance. In vivo data show that BGA002 is able to significantly improve event-free survival in both multidrug-resistant and non-resistant SCLC models. In particular, while the high vascularization characterizing the multidrug-resistant model represented a notable obstacle to be overcome, BGA002 showed the ability to potently reduce the vascularization.

Taken together, our findings on BGA002 open the possibility of improving the efficacy of therapy for patients with MYCN-related SCLC and their prognosis and allows for the opportunity to combine it with immunotherapy and other drugs. Moreover, the safety profile of treatment could also improve for these patients, because BGA002 showed a good toleration profile in preclinical GLP safety regulatory studies in line with the fact that MYCN expression is almost absent in healthy tissues [27]. Furthermore, BGA002 has received orphan drug designation for both SCLC (from the FDA, orphan register: DRU-18-6260) and NB (from the FDA, orphan register: DRU-17-6085 and the European Medicines Agency (EMA), orphan drug application: EMA/OD/020/12), certifying its potential as a novel therapeutic approach for SCLC and NB.

## 5. Conclusions

Different studies have reported the *MYC* family members to be hallmarks for aggressiveness and drug resistance in SCLC. In addition, *MYCN* expression has been recently related to immunotherapy resistance, conferring to this gene a completely new relevance for SCLC. The present study showed that specific *MYCN* inhibition by BGA002 could be a novel viable targeted therapeutic strategy for *MYCN*-related SCLC. In particular, antigene silencing of *MYCN* in SCLC resulted in the block of the mTOR pathway followed by the autophagy activation. Moreover, it was able to revert the pattern expression of specific genes that we found to emerge as a feature for SCLC, such as the upregulation of the anti-inflammatory gene *TNFAIP3*, the downregulation of the oncogenes *CNTFR* and *DLX5*, and, finally, the triggering of apoptosis. These findings occurred also in the context of multidrug resistance, thus paving the way to overcome the most recurrent and rapidly emerging phenomenon in SCLC’s clinical course.

Finally, in vivo treatment with BGA002 increased survival in MNA-SCLC, even in the multidrug resistance scenario, in which it strongly reduced the related tumor vascularization. Because *MYCN* has a very restricted expression in healthy cells, the specific block of *MYCN* expression by BGA002 constitutes a new and promising precision medicine approach.

The results obtained by *MYCN* silencing by BGA002 in SCLC were also similarly found when treating NB, another highly aggressive and *MYCN*-related neuroendocrine tumor. As we had previously found that *MYCN* drives immunosuppression in different tumors and that BGA002 was able to restore immune-response in *MYCN*-related NB, the *MYCN* silencing in SCLC by BGA002 opens the way to new immunotherapeutic approaches for the SCLC-N subtype as well, which is thus far resistant to immunotherapy.

Alterations of *MYCN* are not restricted to SCLC and define a wide range of tumors with poor prognosis, chemoresistance and lack of response to immunotherapies. Therefore, BGA002 treatment could be extended to all *MYCN*-driven tumors.

## Figures and Tables

**Figure 1 cancers-15-00990-f001:**
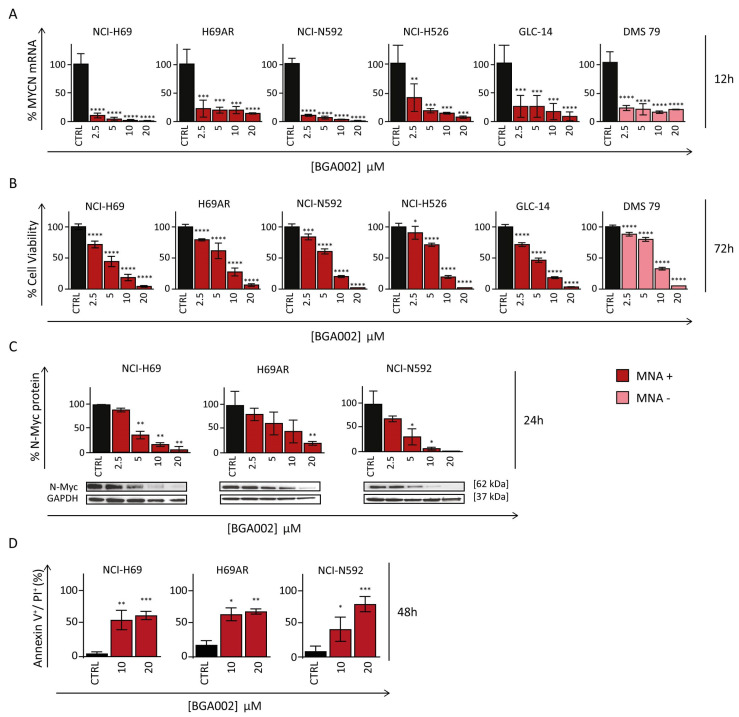
BGA002 reduces *MYCN* expression leading to cell growth inhibition and apoptosis activation in SCLC cell lines. (**A**) MYCN mRNA expression inhibition through RT-PCR after 12 h of treatment in SCLC cell lines. Bars, mean; whiskers, SD (n = 3, biological replicates for each cell line). (**B**) Cell viability assay showing a decrease after 72 h of treatment. Bars, mean; whiskers, SD (n = 3, biological replicates for each cell line). (**C**) Representative Western blot analysis after 24 h of treatment. Representative staining for N-Myc (top) and associated GAPDH staining (bottom) is shown. Top, quantification of N-Myc expression (normalized with GAPDH signal). Bars, mean; whiskers, SD (n = 3, biological replicates for each cell line). The original western blots are shown in Appendix A. (**D**) Apoptosis measured after 48 h of treatment for NCI-N592 (right), H69AR (middle) and NCI-H69 (left) cell lines. Bar plots represent the percentage of cells stained by Annexin V+/PI+. The bars represent the mean, and the whiskers are the standard deviation (n = 3 experiments for each cell line). Data are analyzed with a two-tailed Student’s *t*-test: *, *p* ≤ 0.05; **, *p* ≤ 0.01; ***, *p* ≤ 0.001; ****, *p* ≤ 0.0001; where not indicated, *p*-value > 0.05.

**Figure 2 cancers-15-00990-f002:**
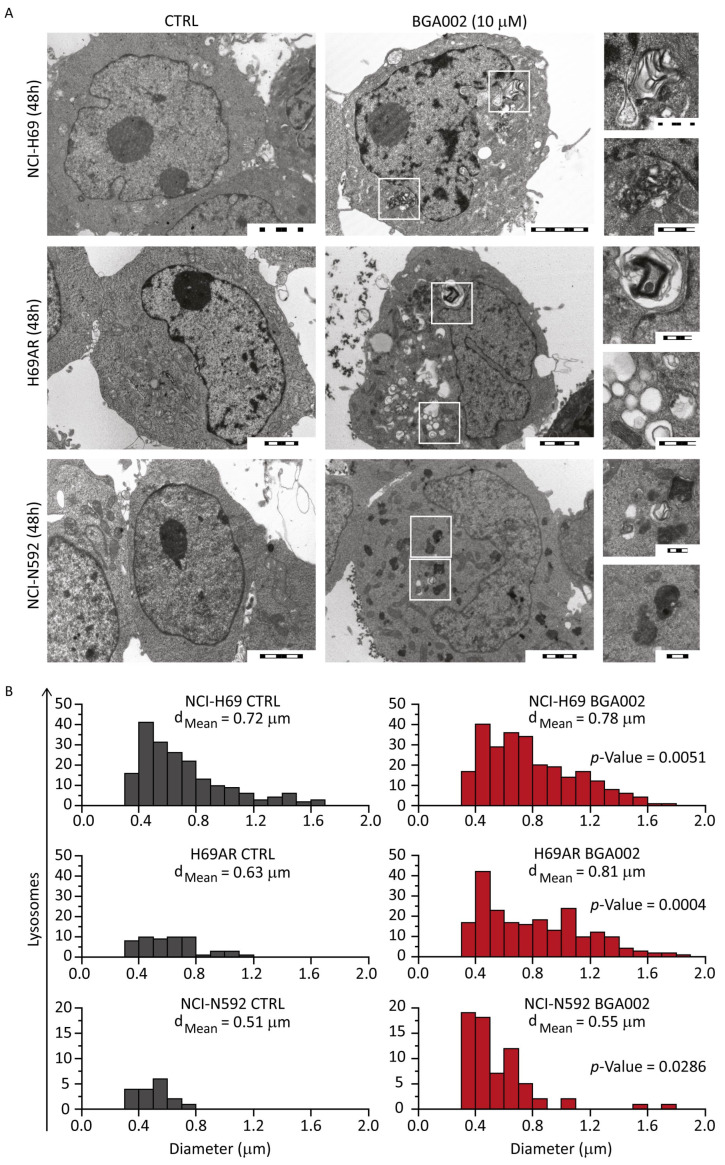
BGA002 modifies lysosome morphological characteristics and distribution in SCLC cell lines. (**A**) Electron microscopy image of SCLC cell lines untreated or treated with BGA002 10 μM for 48 h. Magnification highlights lipid degradation structures and vesicle formations. Scale bar in the whole cell images correspond to 2 μm; scale bar on the magnifications correspond to 500 nm. (**B**) Distribution of lysosome diameters acquired with confocal microscope, in SCLC cell line, untreated (left) or treated (right) with BGA002 for 48 h. Lysosome data for each cell (n = 40) are analyzed with two tailed, paired Student’s *t*-test. Mean lysosome diameter is reported for each cell line.

**Figure 3 cancers-15-00990-f003:**
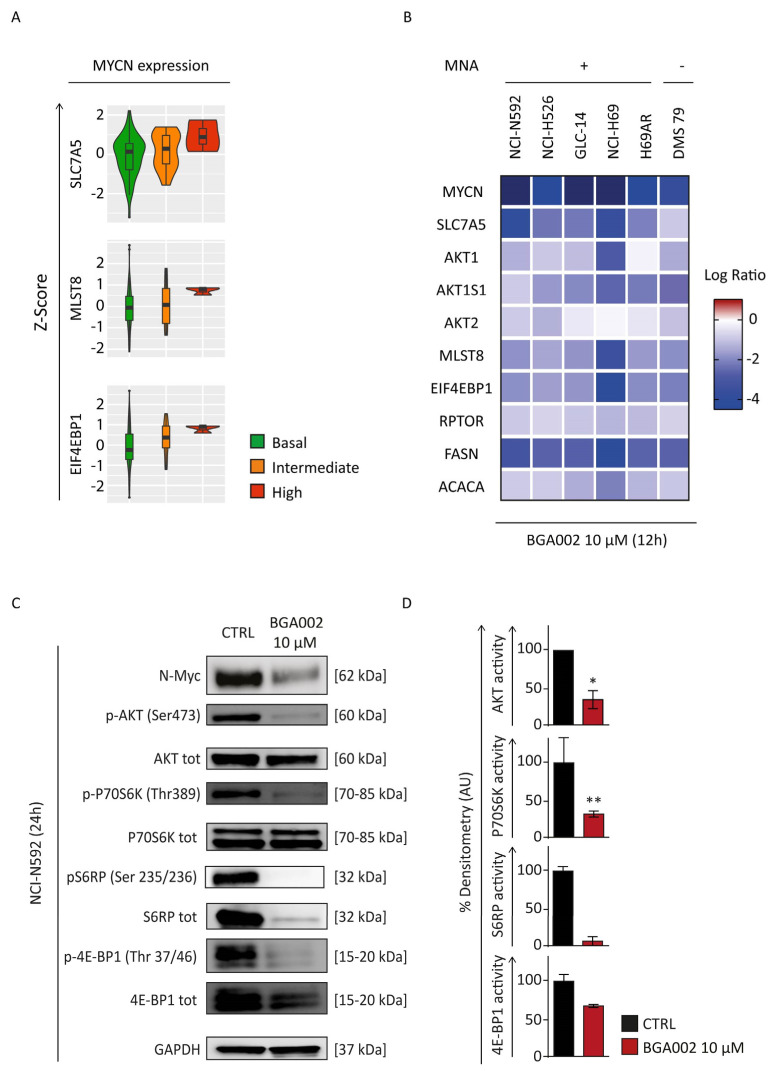
BGA002 leads to mTOR complex inhibition in SCLC cell lines. (**A**) Violin plot reporting *SLC7A5*, *MLST8* and *EIF4EBP1* Z−Score in SCLC patients grouped by MYCN expression level. (**B**) Heat map of the mTOR pathway-related gene expression variation after 12 h of BGA002 treatment (10 µM) in small-cell lung cancer cell lines. Columns represent cell lines, rows represent genes belonging to the mTOR pathway and the color scale represents the log2 fold change in comparison with the untreated cells (n = 3 for each cell line). (**C**) Western blot analysis for mTOR pathway activity in NCI-N592 cell line after 24 h of treatment (representative image of one of the two biological replicates). The original western blots is shown in Appendix A. (**D**) mTOR pathway activity quantification, normalized over the control (n = two experiments). The bars represent the mean, and the whiskers are the standard deviation. Data are analyzed with a two-tailed Student’s *t*-test: *, *p* ≤ 0.05; **, *p* ≤ 0.01; where not indicated *p*-value > 0.05.

**Figure 4 cancers-15-00990-f004:**
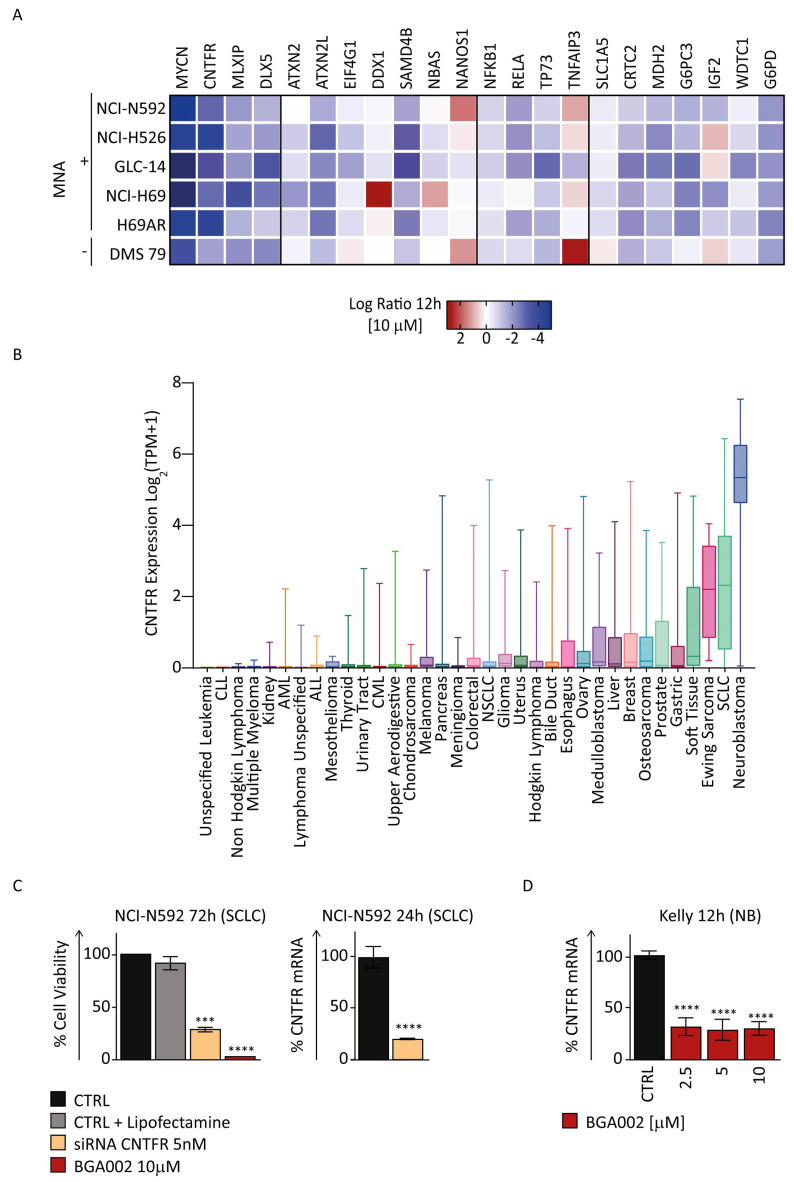
BGA002 regulates the expression of genes involved in cancer progression in SCLC cell lines. (**A**) Heat map of cancer-related gene expression variation after 12 h of BGA002 treatment (10 μM). (**B**) *CNTFR* expression in tumor cell lines as reported in CCLE. SCLC cell lines show the second higher value of *CNTFR* expression among the other tumor type. (**C**) Cell viability assay after treatment with BGA002 or *CNTFR* siRNA in NCI−N592 cell line. Gene expression inhibition in NCI−N592 cell line after treatment with the same *CNTFR* siRNA used for viability assessment. (**D**) *CNTFR* expression inhibition after treatment with BGA002 for 12 h at different concentrations of the MNA−NB cell line. The bars represent the mean, and the whiskers are the standard deviation. Data are analyzed with a two-tailed Student’s *t*-test: ***, *p* ≤ 0.001; ****, *p* ≤ 0.0001; where not indicated *p*−value > 0.05.

**Figure 5 cancers-15-00990-f005:**
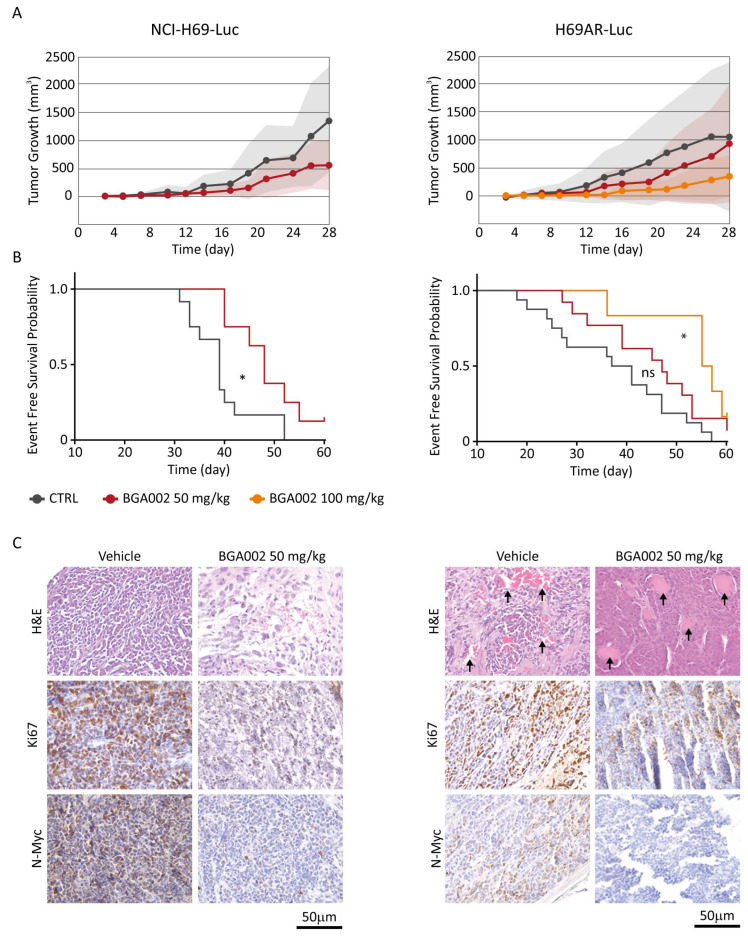
BGA002 reduces tumor growth and increases EFS in SCLC mouse models. (**A**) Evaluation of tumor growth in SCLC xenograft mouse models (NCI-H69-Luc on the left, H69AR-Luc on the right) treated with vehicle (respectively n = 12 and n = 16), BGA002 50 mg/kg/day (respectively n = 8 and n = 13) and BGA002 100 mg/kg/day (n = 6). (**B**) Kaplan–Meier plots for the probability of event-free survival over time for mice (NCI-H69-Luc and H69AR-Luc xenograft) treated with vehicle (black; respectively n = 12 and n = 16), BGA002 50 mg/kg/day (red; respectively n = 8 and n = 13) and BGA002 100 mg/kg/day (orange; n = 6) from the start of treatment. In the middle of the plot is the associated *p*-value (log-rank test) *, *p*-value ≤ 0.05. (**C**) IHC analysis of NCI-H69-Luc and H69AR-Luc mice untreated or treated with BGA002 50 mg/kg/day. Images of sections are shown stained with hematoxylin and eosin (H&E; first row), Ki-67 antibody (second row), N-Myc antibody (third row). Similar results were obtained from four independent mice. Black arrows indicate vascular structures.

## Data Availability

Expression data analyzed in this study were obtained from Cancer Cell Line Encyclopedia, from the European Genome-phenome Archive, under accession number EGAS00001000925 and from EMBL-EB, using accession number E-MTAB-1781.

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
