# Peer review of "Antigene *MYCN* Silencing by BGA002 Inhibits SCLC Progression Blocking mTOR Pathway and Overcomes Multidrug Resistance"

_cancers, 2023, doi:10.3390/cancers15030990_

Round 1

Reviewer 1 Report

Authors show MYCN silencing by BGA002 inhibits SCLC progression and overcomes drug resistance. Below are a few questions authors need to address. 

what is the clinical relevant dose of BGA002 ? because authors show at 10 uM molecular mechanisms are altered. also in xenografts at 50 mg/kg one of the xenograft response was suboptimal. So how relevant are these preclinical studies in terms of the clinic? 

How does BGA002 compare with other MYCN inhibitors?

Reviewer 2 Report

With pleasure, I read the paper titled: “Antigene MYCN silencing by BGA002 inhibits SCLC progression blocking mTOR pathway and overcoming multidrug-resistance”. Overall, the authors examined the efficacy of BGA002 (specific MYCN inhibitor) on SCLC with MYCN amplification and depicted this drug was effective in vitro by reducing viability, inducing apoptosis, activating autophagy, blocking mTOR, and downregulating the protein/mRNA expression of important MYCN downstream genes and other oncogenes. Additionally, the authors showed that monotherapy BGA002 exhibited favorable in vivo anticancer effects on SCLC xenografts, including ones with tumor-resistance. I have a few questions and comments below.

Figure 1. I wonder why the authors reported different time points for their assays (mRNA at 12 hours, protein levels at 24 hours, cell viability at 72 hours, and apoptosis at 48 hours). This may need a clarification. Are there other SCLC cell lines without MNA-amplification beyond DMS 79? If there are any, it would be great if such cell lines can be added. For the non-MYCN amplified cell line DMS 79, did you examine the apoptosis and the protein profile? For panel C, it is recommended to probe for at least 1-2 apoptosis markers (e.g., cleaved caspase-3 or cleaved PARP to validate your apoptosis findings in Panel D). The cell line DMS 79 with MYCN overexpression appeared to exhibit similar responses to the MYCN-amplified SCLC cells. What is the effect of BGA002 on SCLC cells without MYCN amplification or MYCN overexpression? Did you examine the safety of BGA002 on other normal cells (e.g., human fibroblasts like HS68/BJ and importantly normal lung cells)? If you overexpress MYCN in HEK293 cells, will the cells become more sensitive to BGA002 (as a rescue experiment). From Supplemental Figure 1, it does not look clear to me that there was significant difference in EC50 between the cell lines with and without MYCN amplification. Have you looked into the GI50 (cytostatic) and LC50 (cytotoxic) values between these cell lines? Does BGA002 promote differentiation of SCLC cells?

Figure 2. “Lipid degradation, normally associated with autophagy process, has been found only in NCI- 267H69 and H69AR cell lines, but not in NCI-N592”. Any potential hypothesis for lack of effect in NCI-N592 cells based on genetic or clinical information? In order to appreciate the differential effects of BGA002 on MYCN-amplified and non-MYCN-amplified SCLC cells, it is important to include data for DMS 79 cells and other related cells.

Figure 3. The authors previously claimed autophagy would be increased upon BGA002 treatment. I expect this to happen as well since mTORC1 complex appeared to be inhibited upon BGA002. Nonetheless, it would be great if authors could probe for proteins that reflect increased autophagy to support their previous claim (Figure 2). Do expect pharmacolgical inhibition of mTOR plus BGA002 treatment will result in synergistic anticancer effects? Have you done ‘unbiased’ pathway analysis to examine the pathways enriched or depleted upon BGA002 treatment?

Figure 4. The rationale behind panel C does not seem clear to me. Most importantly, have you looked into whether gene or mRNA profile is matched between siRNA knockdown of MYCN and pharmacological inhibition of MYCN with BGA002? Also, if MYCN is genetically depleted in these MYCN-amplified SCLC cells (via siRNA or shRNA or CRISPR/CAS9), does BGA002 lose sensitivity in these cells?

Figure 5. Why 100 mg/kg dose was not used for NCI-H69-Luc xenografts?  Were the differences in tumor growth curves statistically significant (panel A)? In real-world, medical oncologists prefer combination therapy (with standard of care chemotherapeutics) rather than mere novel single-agents. I wonder if you have experimented combining BGA002 with standard of care chemotherapeutics. Have you looked into IHC markers for vascularization (CD31 or CD34) and apoptosis (cleaved caspase-3 to validate your in-vitro results)? Lastly, it would be great if you had harvested xenografts and examined their protein profile for MYCN and other relevant proteins. If you have RNA samples for RNA-seq, this would be a big bonus, too. Did you try BGA002 on DMS 79?

Overall. The manuscript is well-written. The references are up-to-date and appropriate. The manuscript needs polishing for English language.

Reviewer 3 Report

Review for the manuscript ID: Cancers-2183541

Submission Title: Antigene MYCN silencing by BGA002 inhibits SCLC progression blocking mTOR pathway and overcoming multidrug-resistance

Authors: Sonia Bortolotti , Silvia Angelucci , Luca Montemurro , Damiano Bartolucci , Salvatore Raieli , Silvia Lampis , Camilla Amadesi , Annalisa Scardovi , Giammario Nieddu , Lucia Cerisoli , Francesca Paganelli , Francesca Chiarini , Gabriella Teti , Mirella Falconi , Andrea Pession , Patrizia Hrelia , Roberto Tonelli *

An article is dedicated for the investigation of a new approach for the one of the most aggressive form of lung cancer: small cell lung cancer (SCLC). Authors propose an innovative approach of antigene MYCN inhibition by a MYCN-specific antigene PNA oligonucleotide (BGA002) and found that systemic treatment with BGA002 significantly increases survival in MYCN-amplified SCLC mouse models, including in a multidrug-resistant model in which also eliminated the tumor vascularization. These important findings warrant clinical testing of BGA002 in MYCN-related SCLC.

The manuscript is interesting, original, very well written, and only needs some minor technical corrections in the main text and references.

Main text.

required corrections:

line 38: please correct „metastatis“ to „metastasis“;

Some other required general corrections:

In line 506 authors

correctly used Italica font for the latin phrase „in vivo“, thus, please re-check and correct all other Latin phrases

(in vitro, in vivo) in lines: 109, 219, 220, 223, 385, 534;

Mycoplasma (line 116);

line 174: uranyl acetate and lead citrate solutions: concentrations of the salts are undefined, please, insert;

line 187, please correct „radius100 μm“ to „radius 100 μm“ (use interval);

line 366, please correct „pathway, [57,58].“ to „pathway [57,58].“;

lines 390-398, please re-check all these lines, and, please correct „mg/kg/die“ to „mg/kg/day“ as authors correctly used in the next page, lines 407 and 412;

References:

general notes for the improvement of references:

please re-check all your references and use only a generally accepted abbreviations for the journal names,

please do not use all capital letters for the journal names (ref. 51, line 720);

please use point after pages;

please use points in the journal names abbreviations (some good examples are given below):

Bamford, S.; Dawson, E.; Forbes, S.; Clements, J.; Pettett, R.; Dogan, A.; Flanagan, A.; Teague, J.; Futreal, P.A.; Stratton, M.R.; et al. The cosmic (catalogue of somatic mutations in cancer) database and website. Br. J. Cancer 2004, 91, 355–358.

Kim, S.T.; Lim, D.H.; Jang, K.T.; Lim, T.; Lee, J.; Choi, Y.L.; Jang, H.L.; Yi, J.H.; Baek, K.K.; Park, S.H.; et al. Impact of KRAS mutations on clinical outcomes in pancreatic cancer patients treated with first-line gemcitabine-based chemotherapy. Mol. Cancer Ther. 2011, 10, 1993–1999.

Calhoun, E.S.; Jones, J.B.; Ashfaq, R.; Adsay, V.; Baker, S.J.; Valentine, V.; Hempen, P.M.; Hilgers, W.; Yeo, C.J.; Hruban, R.H.; et al. BRAF and FBXW7 (CDC4, FBW7, AGO, SEL10) mutations in distinct subsets of pancreatic cancer: Potential therapeutic targets. Am. J. Pathol. 2003, 163, 1255–1260.

----------------------------------------------------------------------------------------------------------------------------

Ref. 75:

Please improve and correct this reference to: „Front. Immunol. 2019, 10, 1084. doi: 10.3389/fimmu.2019.01084“ (add DOI).

After all listed minor corrections will be done, the current manuscript can be accepted for publication.

Round 2

Reviewer 1 Report

NA

Reviewer 2 Report

The authors did a great job by responding to all comments adequately and supporting their thought processes. The manuscript reads very well, intellectually solid, and scientifically sound. I confidently support the manuscript to be accepted in its current form. Congratulations and well done!